# Mechanical Properties of Tensile Cracking in Indium Tin Oxide Films on Polycarbonate Substrates

**Jiali Zhou [1,2], Xuan Zhang [1,2,*], Xiaofeng Zhang [1,2,*], Wenqiao Zhang [1,2], Jiuyong Li [1,2], Yuandong Chen [1,2], Hongyan Liu [1,2] and Yue Yan [1,2,*]**

[1] Beijing Institute of Aeronautical Materials, Beijing 100095, China; z1322463793@163.com (J.Z.); wqz_sunny@163.com (W.Z.); 15737936609@163.com (J.L.); nishiyuandong@163.com (Y.C.); homyeeliu@126.com (H.L.)

[2] Beijing Engineering Research Center of Advanced Structural Transparencies to the Modern Traffic System, Beijing Institute of Aeronautical Materials, Beijing 100095, China

\* Correspondence: kaixuan1226@163.com (X.Z.); flexzhang@126.com (X.Z.); yue.yan@biam.ac.cn (Y.Y.)

**Abstract:** The electro-mechanical behaviors of transparent conductive oxide film on polymer substrate are of great concern because they would greatly affect the stability and lifespan of the corresponding devices. In this paper, indium tin oxide (ITO) films with different thicknesses were deposited on a polycarbonate (PC) sheet; meanwhile, in situ electrical resistance, in situ scanning electron microscopy and profilometry were employed to record the electrical resistance, morphologies and residual stress in order to investigate the fracture behavior and electrical-mechanical properties of ITO films under uniaxial tension loading. The electrical resistance changes, crack initiation, crack propagation and crack density evolution of ITO films were systematically characterized by in situ tests. Three fracture stages of ITO films were summarized: I crack initiation, II crack propagation, III crack saturation and delamination. The crack initiation and electrical failure in a thinner ITO film occurred at relatively higher applied tensile strain; namely, the ductility of the film decreased as the film thickness increased. Residual compressive stress was recorded in the ITO films deposited on PC at room temperature and increased as the film thickness increased. Intrinsic crack initiation strain (CIS*) showed an opposite thickness dependence to residual strain ($\varepsilon_r$); the increase in residual compressive strain was counteracted by the decrease of intrinsic cohesion, leading to an overall decrease in effective crack initiation strain (CIS) when the film thickness increased. In addition, integrated with a formulated mechanics model and the analysis of the three fracture stages under tension, the fracture toughness and interfacial shear strength were quantitatively determined. As the film thickness increased (in the range of 50~500 nm), the fracture toughness decreased and the films were more prone to crack, whereas the interfacial shear strength increased and the films were less likely to delaminate.

**Keywords:** indium tin oxide; polymer substrate; mechanical properties; in situ tests

## 1. Introduction

Indium tin oxide ($In_2O_3$: Sn, ITO) with high electrical conductivity and visible light transmittance is a commonly used transparent conductive film which has important applications and research value in industrial and high-tech fields [1,2]. Inorganic glass has gradually been replaced by polymers with favorable lightness and flexibility as the base material of ITO films. ITO films deposited on polymer substrates are a key material in the development of solar cells [3], electrodes [4], electronic displays [5] and aeronautical transparencies [6].

In the field of aeronautical transparencies, aircraft canopy is the key functional structural component of advanced fighters, usually made of centimeter-thick hard polymer transparent materials [7]. Compared to polymethyl methacrylate, polycarbonate (PC) with higher impact resistance, heat resistance and good optical properties can better meet the high Mach number and bird impact resistance of the aircraft canopy. PC has important

application prospects and values which will become one of the main application materials of aeronautical transparencies [8]. ITO films deposited on the PC surface are an important method to achieve radar stealth [9], and its mechanical durability is a primary issue at present. The PC-ITO is subjected to tensile, bending and other external stress in the course of use, resulting in cracking or interfacial delamination of the ITO film, which may affect the mechanical integrity and optical/electrical properties, thus affecting the function of the device.

Most mechanical studies focus on flexible electronic devices whose substrates are usually micron-thick polyethylene terephthalate (PET) and polyimide (PI) [10], whereas there are few mechanical studies about ITO films on centimeter-thick hard PC substrates in aeronautical transparencies. The thinner flexible polymer substrate typically has large thermal/mechanical mismatches with brittle films; nonetheless, the deformation of brittle films could be retarded through elastoplastic deformation of thinner flexible substrates. Thus, ITO films on the thin flexible polymer substrates had considerable ductility; for instance, cracking was observed in a 105 nm thick ITO film on 120 μm PET at a strain of ~2% [11] and for an 80 nm thick ITO film on 12.7 μm PI at ~1.6% [12]. More serious mechanical failure problems of ITO films on centimeter-thick hard polymer substrates arose in our experiments; for instance, the crack initiation strain (CIS) of a 100 nm thick ITO film on a 0.3 cm PC was only ~1%. In addition, the mechanical reliability of film systems is dependent on the film thickness [13]. It can be found that there are differences in the thickness dependence of the mechanical behavior of different film-substrate systems [14,15]. It is necessary to carry out specific research on mechanical properties and fracture modality of ITO films with different thicknesses on centimeter-thick hard polymer substrates.

In this paper, ITO films with different thicknesses ranging between 50 nm and 500 nm were deposited on centimeter-thick hard PC; then the fracture behavior and mechanism of ITO under tension were studied. Existing studies on the brittle fracture of film on polymer substrates include in situ tests in which the crack density and electrical resistance evolution as a function of the applied strain is monitored [16–18]. The cohesive toughness (which controls cracking) and the adhesive toughness (which controls delamination) of the films on polymer substrates can be estimated based on the experimental data from fragmentation tests [19]. The in situ test can observe the cracks under applied load in time and intuitively is of particular importance to study fracture behavior, which can avoid the unloading-induced partial or full closure of cracks in the ITO thin film in ex-situ tests: The crack initiation and propagation of ITO films under applied load were observed inside SEM; simultaneously, the evolution of crack initiation strain (CIS) and crack density (CD) were determined; the corresponding resistance changes were measured with an electrochemical workstation during the tensile test; then the critical strain for electrical failure ($\varepsilon_c$) was obtained. Simultaneous in situ electrical and SEM tests could not only obtain CIS and $\varepsilon_c$ values reflecting the ductility of the film, but also correlate crack initiation and propagation with its resistance change for exploring the effect of film thickness on mechanical-electrical properties. Integrated with a formulated mechanics model and the analysis of the three fracture stages under tension, the fracture toughness and interfacial shear strength were quantitatively determined to study the effect of film thickness on fracture behavior.

## 2. Materials and Methods

The PC substrates (3 mm thickness, LEXAN 9030, Sabic, Riyadh, Saudi Arabia) were rinsed with neutral detergent and distilled water, followed by drying at 75 °C for 40 min before film deposition. ITO films ($In_2O_3$ and $SnO_2$, weight ratio of 90%:10%, $\Phi$ 71 × 4 mm, purity of 99.99%) with different thicknesses were deposited on PC substrates by means of direct current magnetron sputtering at room temperature. The sputtering power was maintained at 100 W. Argon and oxygen gas with purity 99.999% were introduced into the chamber at a flow rate of 30 and 2 sccm (standard-state cubic centimeter per minute), respectively. The base and working pressures were ~3.0 × $10^{-3}$ Pa and ~0.39 Pa, respectively.

The deposition rate was kept at ~10 nm·min$^{-1}$. ITO films with different thicknesses were obtained by adjusting the sputtering time.

The thickness of ITO film was measured by profilometer, and the sheet resistance was tested by a four-probe tester. The Elastic Modulus of the ITO film was measured through the Tribo-Indenter system (TI 950, Hysitron, Minneapolis, MN, USA) in nanoindentation mode using a Berkovich indenter. The elastic modulus and Poisson's ratio of the PC substrate were determined by tensile testing. The residual stress ($\sigma_r$) of ITO film was calculated from the radii of curvature of the film before, $R_1$, and after, $R_2$, film deposition, following the analysis of substrate curvature method by the profilometer (P-7, KLA-Tencor, San Francisco, CA, USA) based on Stoney's formula:

$$\sigma = \frac{E_s t_s^2}{6(1 - \nu_s)t_f}\left(\frac{1}{R_2} - \frac{1}{R_1}\right) \tag{1}$$

where $E_s$ is the elastic modulus of the substrate, $\nu_s$ is the Poisson's ratio of the substrate, and $t_s$ and $t_f$ are the thickness of substrate and film. $E_s$ and $\nu_s$ were determined as 2.37 GPa and 0.37, respectively.

The critical strain and toughness could be calculated from in situ electrical resistance and SEM test. The strain applied on the ITO films which deposited on the PC substrates were equivalently recorded as the one of PC-ITO substrate-film systems because they were attached firmly. There were two kinds of in situ measurement. The first (hereby named Method-I) was an in situ electrical resistance test under axial tension loading via a universal mechanical test. A uniaxial tensile test was carried out with the specimen dimensions (50 mm in gauge length and 10 mm in width), and the strain rate was fixed at 1 mm·min$^{-1}$. During the tensile process, the electrical resistance of the sample was measured by an electrochemical workstation (660, CHI) in the multi-potential step mode to obtain the resistance change $\Delta R/R$ versus strain curve, and thus the value of $\varepsilon_c$. Additionally, other strain rates (3 mm·min$^{-1}$ and 5 mm·min$^{-1}$) were applied to investigate strain rate dependence. The surface morphology of the samples after the axial tension was recorded ex situ by a confocal laser scanning microscope (OLS500S, OLYMPUS, Tokyo, Japan) to characterize the cracks. The second (hereby named Method-II) was the in situ SEM test under tension. A microtester (MINI-MTS4000, Qiyue Technology Co., Ltd., Beijing, China) inside the high-resolution field emission SEM chamber (TESCAN 8000, TESCAN, Brno, Czech Republic) was employed. The maximum force capacity of the microtester was 4000 N with a resolution of 0.1 N. The dimensions of the sample for in situ SEM testing were 26 mm in gauge length and 3 mm in width. The uniaxial tensile tests were conducted under displacement control, and the strain rate was controlled to be around 0.5 μm·s$^{-1}$. The increase of applied tensile strain was paused at certain preset strain values to allow for high resolution SEM imaging at different locations along the sample length, recording of crack development via a CCD camera attached to the SEM and determination of crack initiation strain and crack density.

## 3. Results

### 3.1. Thickness and Strain Rate Dependence of Film Failure with Method-I

The electrical resistance variation of the ITO films with increasing applied tensile strain is shown in Figure 1. If the critical strain when the initial crack occurs, $\varepsilon_c$, was defined as the 20% increase of electrical resistance compared to the original value $R_0$, as indicated in Figure 1a, $\varepsilon_c$ of ITO films decreased from 1.17% to 0.66% with the film thickness increased from 50 nm to 500 nm. For each thickness of ITO films, the resistance remains nearly unchanged when the applied tensile strain was relatively small. As the applied tensile strain was further increased, the ITO electrical resistance elevated dramatically when the tensile strain approached the value at the electrical failure strain. In addition, the ductility of the ITO films with the same thickness was slightly improved with decreasing strain rate, which might be due to the increase of the elastic modulus caused by the increasing strain rate during the test [16], but there was little correlation, as shown in Figure 1b. Therefore, the strain rate was fixed at 1 mm·min$^{-1}$.

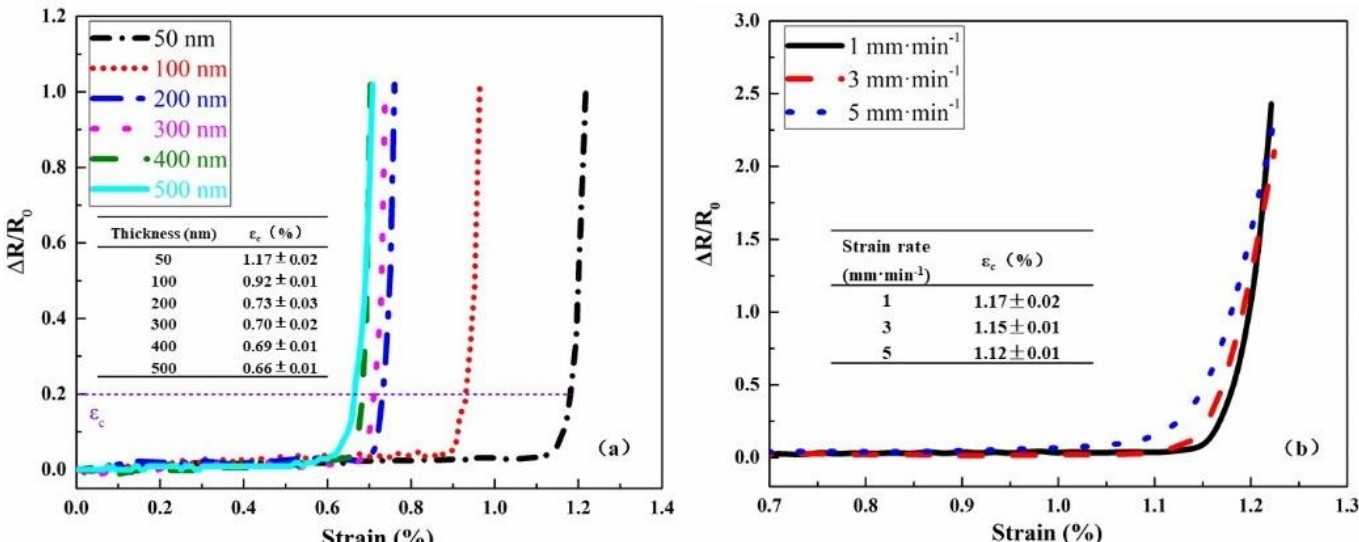

**Figure 1.** (**a**) Fractional change in resistance ($\Delta R/R_0$) of ITO with different thicknesses deposited on PC as a function of strain $\varepsilon$; (**b**) $\Delta R/R_0 \sim \varepsilon$ curve of ITO with 500 nm thickness deposited on PC under different strain rates. $R_0$ denoted the electrical resistance of the unloaded ITO thin film.

The surface morphology of the tested ITO films after unloading was characterized using a confocal laser scanning microscope. As shown in the illustrations in Figure 2, no significant cracks were observed in the samples on which the external tensile strain was up to 2.2%, but crack morphology could be observed at even higher 5% tensile strain. The plastic deformation of the substrate was small at 2.2% strain, and the deformation after unloading basically returned, leading to the occlusion of the crack. Thus, in situ tests had to be implemented to avoid the unloading-induced partial or full closure of cracks in the ITO thin film in ex-situ tests.

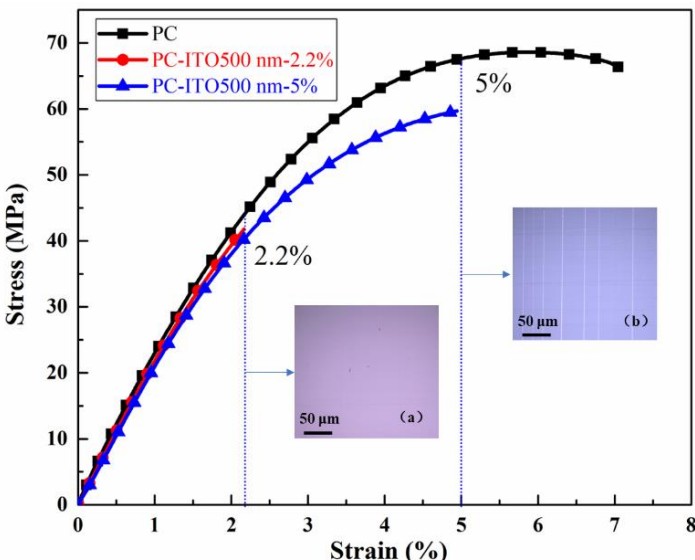

**Figure 2.** Stress–strain curves of PC substrate and ITO film on PC. Illustrations are surface topography of 500 nm ITO film after unloading: (**a**) Stretched to 2.2% strain; (**b**) stretched to 5% strain.

### 3.2. Thickness and Strain Rate Dependence of Film Failure with Method-II

Three fracture stages of 50 nm-thickness ITO film under uniaxial tension were summarized from in situ SEM tests, as shown in Figure 3. Stage I: Crack initiation. Vertical cracks developed in a normal direction of the loading initiated randomly in the film and started propagating at a critical strain referred to as CIS. Stage II: Crack propagation. Vertical

cracks propagated upward and downward perpendicular to the tensile direction as the strain increased, and the number of cracks in the unit sample length increased rapidly along the tensile direction. The crack density could be defined as the number of vertical cracks per unit length in the direction of tensile load, CD (mm$^{-1}$). The generation of vertical cracks diminished and transverse cracks may be observed across fragments due to Poisson's ratio effects. Stage III: Crack saturation and delamination. No further vertical cracks were generated in this stage and the density of cracks reached a saturation value, CD$_s$, related to the intrinsic properties of the material system. Transverse buckling was observed across fragments due to a mismatch in Poisson's ratio [20] and/or stress relaxation in the tensile direction [21], and then delamination occurred, which became the dominant failure mechanism.

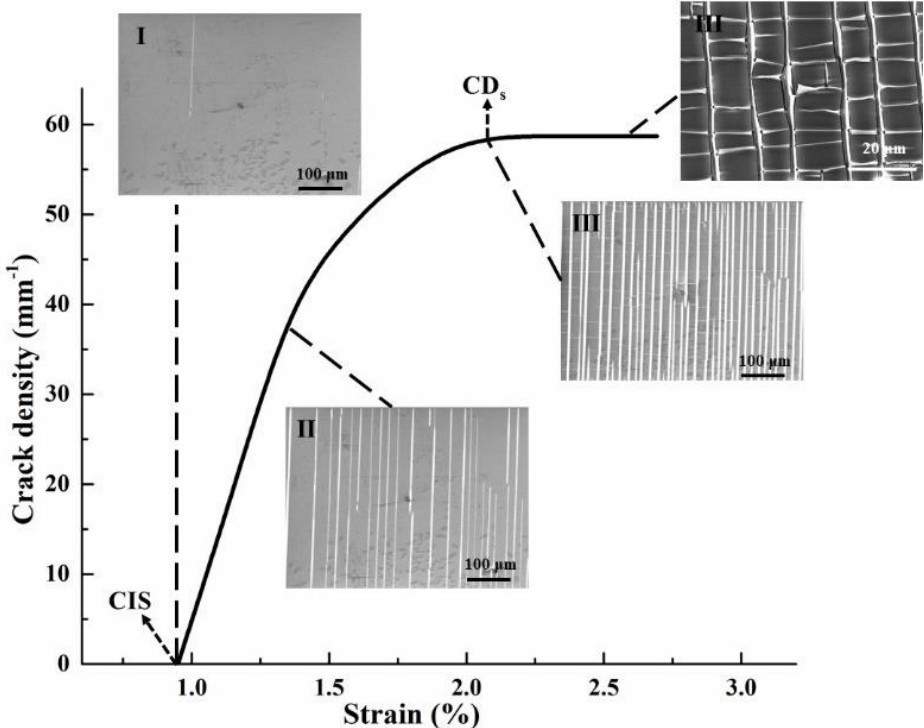

**Figure 3.** Three fracture stages of ITO film (50 nm) under uniaxial tension: Stage I, crack initiation; stage II, crack propagation; stage III, crack saturation and delamination.

　　The typical process of crack initiation in stage I is illustrated in Figure 4a,b. There were no appreciable cracks before loading. As the tensile strain increased to a threshold value 0.95% (CIS was 0.95% for 50 nm thick ITO film), the first vertical crack occurred at random locations in the ITO film. The crack propagation process in stage II is shown in Figure 4c–e. The number of vertical cracks along the tensile direction increased rapidly as the tensile strain increased, i.e., CD increased. A crack tip was found near the top edge of the blue solid rectangle box in Figure 4c at the tensile strain of 1.15% and this crack propagated upward and advanced outside of the same blue solid rectangle box at the tensile strain of 1.35% (Figure 4d). Meanwhile, cracks not existing at 1.15% strain in Figure 4c appeared in the red dash rectangular box at 1.35% strain in Figure 4d, and continued to grow upward at 1.54% strain (Figure 4e). The process of crack saturation and delamination in stage III is demonstrated in Figure 4f,g. As the strain increased further, the generation of vertical cracks diminished and crack saturation was reached; simultaneously, the lateral shrinkage of the film layer caused the generation of transverse cracks due to the Poisson's ratio effects (Figure 4g). With further increasing strain, the increased transverse compressive stress drive resulted in the increase of the number of transverse cracks and the occurrence of buckling, followed by fracture and delamination (Figure 4h,i).

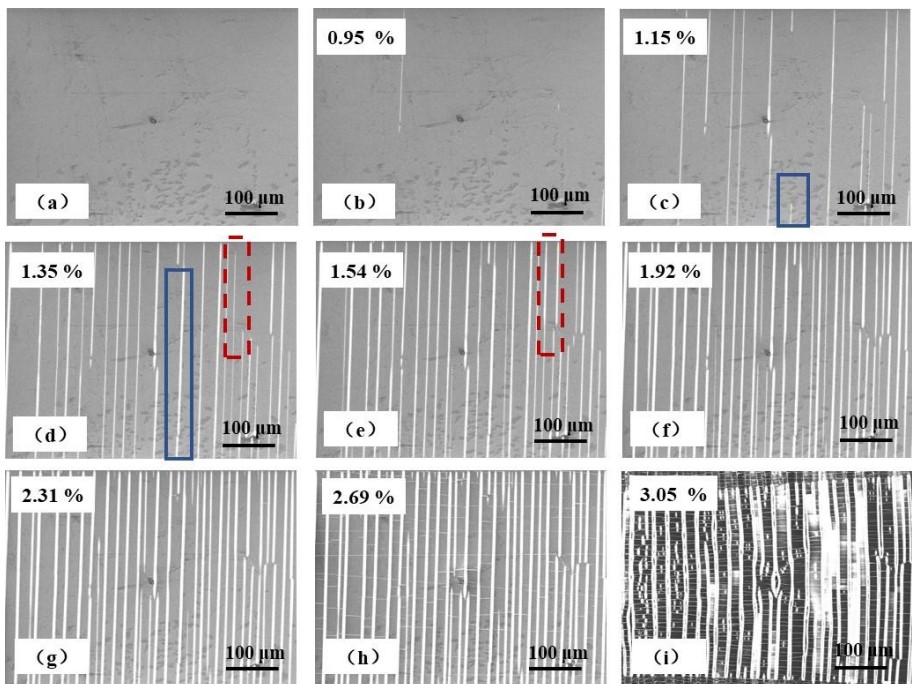

**Figure 4.** Typical process of crack initiation, propagation, saturation and delamination in 50 nm ITO film on PC. (**a**,**b**) were stage I, crack initiation; (**c**–**e**) were stage II, crack propagation; (**f**–**i**) were stage III, crack saturation and delamination.

Buckling could be clearly seen, as shown in Figure 5. Buckling would cause the bending of the ITO fragments, resulting in tensile stress concentration at the apex of the bulge. Eventually the ITO film fractured parallel to the tensile direction and resulted in delamination (Figure 5b). With further lateral shrinkage, the crack fragments tended to overlap (red solid rectangular box in Figure 5b), which further contributed to the delamination.

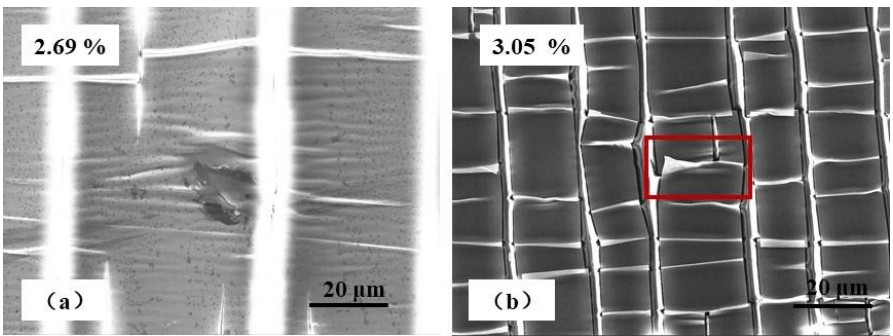

**Figure 5.** Buckling and fracture of 50 nm ITO film on PC. (**b**) was an enlarged drawing of (**a**).

### 3.3. Thickness Dependence of Crack Initiation Strain and Crack Density Evolution

Crack density versus strain curve for different thicknesses of ITO films are displayed in Figure 6a; the curves of CIS and $CD_s$ versus film thickness are shown in Figure 6b. For each thickness of ITO film, CD increased significantly with tensile strain increased and saturated gradually. CIS and $CD_s$ decreased as the film thickness increased, i.e., the crack initiation in a thicker ITO film occurred at relatively smaller applied tensile strain than that for a thinner ITO film. However, the saturated crack density of the thicker ITO film was lower than that of the thinner ITO film. The 500 nm thick ITO film possessed the smallest crack initiation strain (CIS of 0.44%) and the lowest crack saturation density ($CD_s$ of 32 mm$^{-1}$).

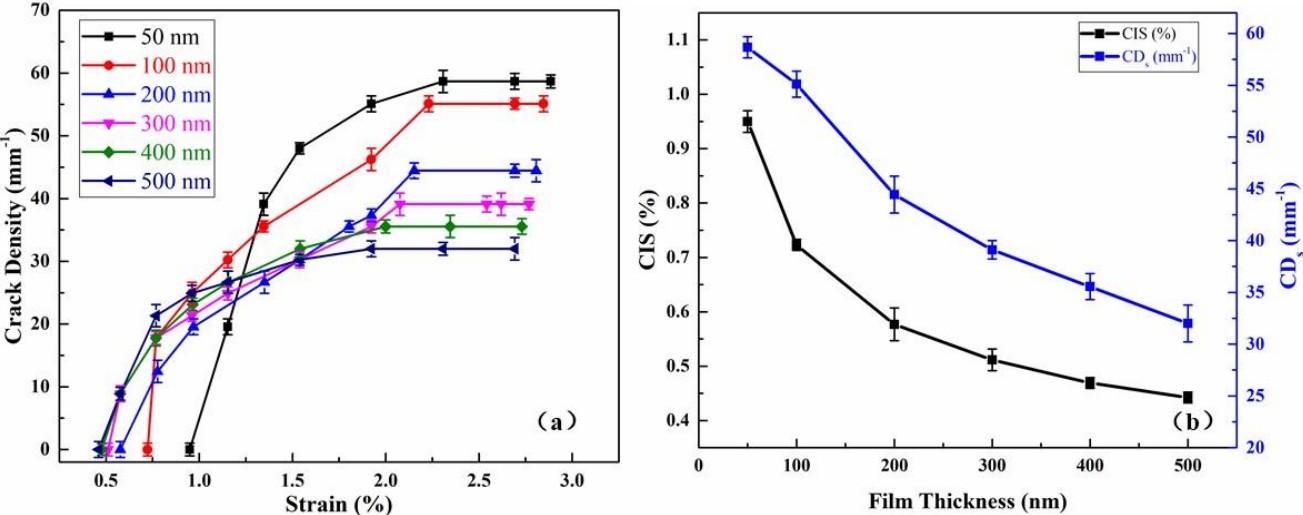

**Figure 6.** ITO films with different thicknesses on PC substrate: (**a**) CD; (**b**) CIS(t) and $CD_s$ (t).

## 4. Discussion

### 4.1. Mechanical-Electrical Properties of ITO Films

The crack density and resistance variation of ITO films with different thicknesses on PC substrates with the applied increasing tensile strain is shown in Figure 7a. The electrical resistance of the ITO films remained nearly unchanged when the applied tensile strain was relatively small. With the further increase of tensile strain, a sudden increase of resistance occurred, whereas the crack density gradually increased. Such a correlation between the evolution of crack density and the change in electrical resistance of the ITO films under tension could be explained from the percolation theory as follows.

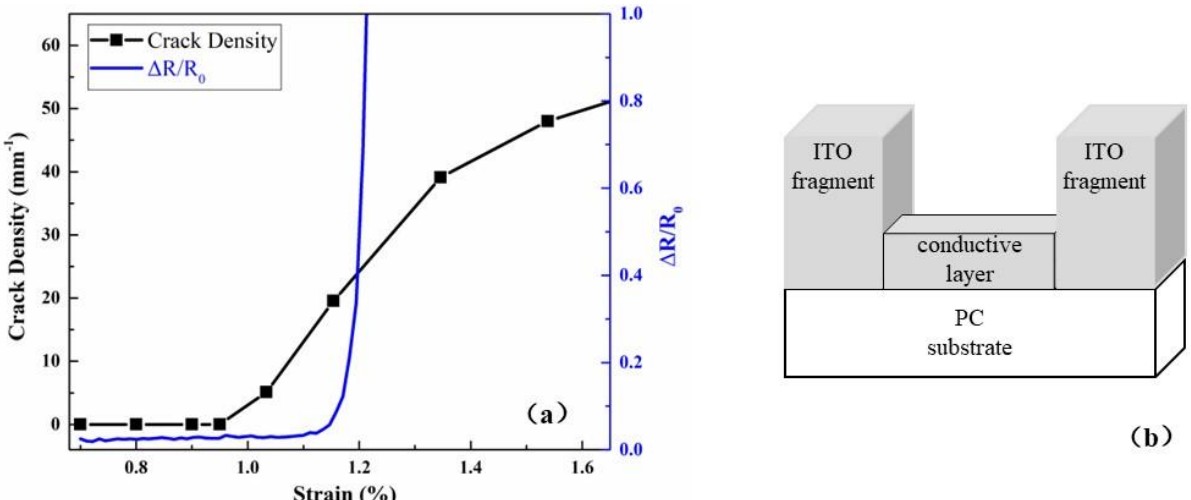

**Figure 7.** (**a**) The crack density and the variation of electrical resistance of ITO films with different thicknesses deposited on PC substrates as a function of the applied strain; (**b**) A model showing the conductive layer between ITO fragments.

At the crack initiation, cracks started to nucleate but did not fully propagate throughout the whole sample width and thickness where a conductive layer existed between ITO fragments, as shown in Figure 7b. As the current could percolate through the whole ITO films because of the conductive layer, the resistivity of ITO films was almost unchanged. As the applied tensile strain increased modestly, crack density increased and cracks started to propagate through the width and thickness of the samples, and the conductive layer thereupon decreased (i.e., the volume fraction of conductive particles decreased). Here

the resistivity increased slowly, which was not proportional with the volume fraction of conductive particles. As the applied strain further increased, the conductive layer further decreased until the resistivity increased sharply at the critical strain "$\varepsilon_c$".

### 4.2. Stress of ITO Films

The influence of film thickness on $\sigma_r$ and CIS and intrinsic crack initiation strain (CIS*) for ITO films is depicted in Figure 8. As listed in Table 1, residual compressive stress was recorded in the ITO thin films and increased as the film thickness increased. Amorphous ITO film deposited at room temperature on glass was also reported to be under compressive stress [22]. Generally, stress in thin films is caused by either the difference in thermal expansion of substrate and film material (thermal stress) or from the microstructure of deposited film (intrinsic stress) [23].

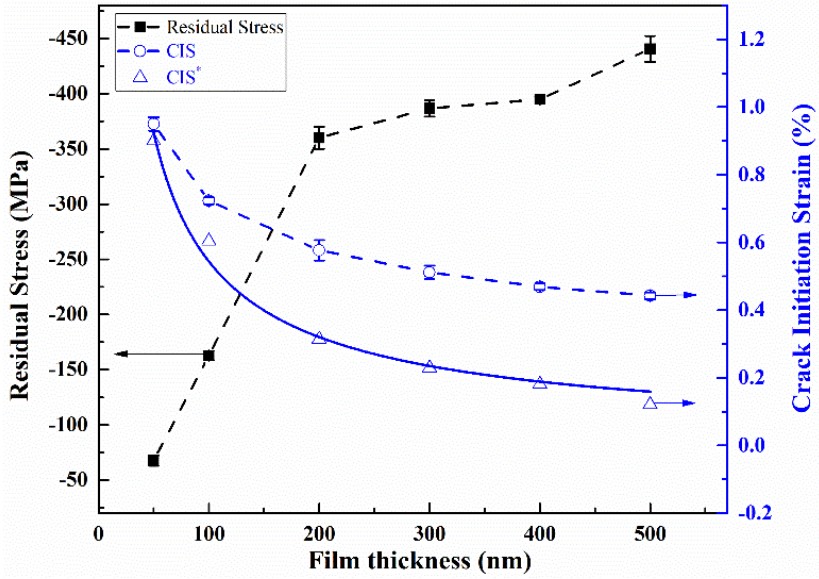

**Figure 8.** Film thickness dependence of internal stress measured crack initiation strain, and intrinsic crack initiation strain of ITO films. The thick continuous lines are power-law fits (with exponent equal to −0.76) to CIS*.

**Table 1.** $\sigma_r$ and CIS and CIS* of ITO films with different thicknesses.

| ITO Thickness (nm) | $\sigma_r$ (MPa) | CIS (%) | CIS* (%) |
|---|---|---|---|
| 50 | −67.57 ± 4.71 | 0.95 ± 0.02 | 0.90 |
| 100 | −162.95 ± 4.17 | 0.72 ± 0.01 | 0.60 |
| 200 | −360.10 ± 10.07 | 0.58 ± 0.03 | 0.31 |
| 300 | −386.725 ± 7.39 | 0.51 ± 0.02 | 0.23 |
| 400 | −394.91 ± 2.78 | 0.47 ± 0.01 | 0.18 |
| 500 | −440.62 ± 11.75 | 0.44 ± 0.01 | 0.12 |

The results of in situ tests show that both CIS and CIS* decreased with the film thickness increased, as listed in Table 2. The crack initiation strain (CIS) was defined as:

$$CIS = CIS^* - \varepsilon_r \tag{2}$$

where $\varepsilon_r = \sigma_r/E_f$ was residual strain. In theory, residual compressive strain (negative by convention) was beneficial since it increased the net strain for tensile failure of the film. However, similar results that CIS increased with increased residual compressive strain could not be obtained in our experiments. Since CIS was a linear combination of CIS* and residual strain (Equation (2)), its thickness dependence resulted from the

thickness dependence of the intrinsic cohesive properties (CIS*) and of the residual strain. The thickness dependence of the cohesive properties (CIS*) could be explained based on fracture mechanics and assuming a linear elastic behavior: $CIS \propto t_f^{-1/2}$ [23]. The power-law fit was performed on the CIS* obtained according to Equation (2) to obtain the CIS*-$t_f$ solid continuous curve with exponent equal to −0.7 as shown in Figure 8. In the range of crack initiation strain, the film-substrate system was basically in the linear elastic stage; thus the linear elasticity assumption was reasonable. The reason why the reported strain values were systematically too high could be attributed to the compliance of the testing machine, and additional slippage effects of the sample in the clamps not being corrected for. $\sigma_r$ increased as the film thickness increased in our experiment, i.e., $\varepsilon_r$ increased as the film thickness increased. In a word, CIS* showed an opposite thickness dependence to $\varepsilon_r$; the increase in compressive strain was counteracted by the decrease of intrinsic cohesion, leading to an overall decrease in effective CIS with the film thickness increased.

**Table 2.** The $G_c$, $K_c$, IFSS and IFSS* values of different samples.

| ITO Thickness (nm) | $G_c$ (J/m$^2$) | $K_c$ (MPa·m$^{1/2}$) | IFSS (MPa) | IFSS* (MPa) |
|:---:|:---:|:---:|:---:|:---:|
| 50 | 10.45 | 1.20 | 5.10 | 4.93 |
| 100 | 9.41 | 1.14 | 7.29 | 6.49 |
| 200 | 5.08 | 0.83 | 9.39 | 6.53 |
| 300 | 4.06 | 0.75 | 10.99 | 6.94 |
| 400 | 3.37 | 0.68 | 12.22 | 7.20 |
| 500 | 1.88 | 0.51 | 12.89 | 6.59 |

*4.3. The Fracture Toughness and Interfacial Shear Strength*

4.3.1. Fracture Toughness

The stress intensity factor ($K_c$) and the critical strain energy release rate ($G_c$) were commonly used to describe the fracture toughness of the film upon cracking. Theoretically, the larger the $K_c$ and $G_c$, the greater the fracture toughness, namely, the less susceptible the material would be to fracturing. $G_c$, for the film system under linear elastic plane strain under uniaxial tension [24,25], was defined as:

$$G_c = \frac{1}{2}\overline{E}_f \varepsilon_i^2 \pi t_f g(\alpha, \beta) \tag{3}$$

where $\overline{E}_f = E_f/(1 - v_f^2)$ was the plane strain modulus of the film, $g(\alpha, \beta)$ was a non-dimensional integral of the crack opening displacement which depended on the elastic mismatch between film and substrate. $g(\alpha, \beta)$ was a function of the Dundurs parameters $\alpha$ and $\beta$, and can be found in: $\alpha = \frac{\overline{E}_f - \overline{E}_s}{\overline{E}_f + \overline{E}_s}$, $\beta = \frac{\mu_f(1-2v_s)-\mu_s(1-2v_f)}{2\mu_f(1-v_s)+2\mu_s(1-v_f)}$, where $\overline{E}_s = E_s/(1 - v_s^2)$ was the plane strain modulus of the substrate, $\mu_f = E_f/(2 + 2v_f)$ and $\mu_s = E_s/(2 + 2v_s)$ were the shear moduli of the film and the substrate, respectively. $v_f$ and $v_s$ were the Poisson's ratio of the film and the substrate, respectively. If $\sigma_r$ in the film was considered, $\varepsilon_i$ (i.e., CIS*) was the sum of CIS and $\varepsilon_r$, $CIS^* = CIS + \varepsilon_r$. Accordingly, $G_c$ could be expressed as:

$$G_c = \frac{1}{2}\pi g(\alpha, \beta)\overline{E}_f \left(CIS + \frac{\sigma_r}{E_f}\right)^2 t_f \tag{4}$$

It could be seen from Equation (4) that $G_c$ was not only a strong function of the elastic mismatch between film and the substrate, but also closely related to the film thickness and residual stress. In addition, the energy release rate $G_c$ was related to the stress intensity factor $K_c$ of the crack by Equation (5):

$$G_c = \frac{K_c^2}{E} \tag{5}$$

The elastic modulus of the ITO film was 137 GPa which was taken from values obtained on a Si substrate using nanoindentation tests, and its Poisson's ratio was 0.25, taken from the literature [26]. In our experiment, $\alpha$ was about 0.963, and in the present work we used $\beta = \alpha/4$. According to Equations (4) and (5), the $G_c$ and $K_c$ values of different samples were calculated, and are listed in Table 2. As the film thickness increased from 50 nm to 500 nm, $G_c$ decreased from 10.45 J/m$^2$ to 1.88 J/m$^2$ and K decreased from 1.20 MPa·m$^{1/2}$ to 0.51 MPa·m$^{1/2}$. The above-mentioned results indicate that the fracture toughness decreased with increased film thickness, and the film was more prone to fracture.

### 4.3.2. The Interfacial Shear Strength

It has been shown that the adhesion of the thin films to the substrate is an important property [27]. In poor adhesion, it would accelerate delamination and fracture, causing a device failure. In the case of good adhesion, the film remains connected to the substrate and it would slow down delamination to ensure that the mechanical and electrical integrity are intact [28]. The adhesion of the ITO films deposited on PC substrates was qualitatively characterized by cross-cut and tape peel tests (ASTMD 3359). No obvious differences in adhesion of ITO films with different thicknesses were found and all were ranked as 5B.

As strain increased, transverse buckling and delamination were observed across fragments due to a mismatch in Poisson's ratio and/or stress relaxation in the tensile direction. Delamination was determined by adhesive properties (interfacial shear strength) that were derived from the analysis of the buckling in stage II and delamination in stage III [29]. The Kelly–Tyson approach for interface shear strength calculation was applicable to brittle film/ductile substrate quantitative studies of adhesion [30]. Microcracks appear when the film was subjected to tensile stress ($\sigma_f$), and the films were divided into many segments. The tensile stress in the film originated from two aspects, which existed through a transfer across $\tau$ at the interface and the residual compressive stress $\sigma_r$ existed. As the tensile strain increased, new microcracks were continuously generated. However, this process would stop when $\sigma_f$ in the film fragment was released, and a saturation region occurred. When $\tau$ increased to $\tau_{max}$, the film debonded from the substrate and delaminated. The interfacial shear strength (IFSS) was equal to $\tau_{max}$ and could be calculated from the analysis of the saturation region III, shown in Figure 5:

$$IFSS = 1.337 t_f E_f CISCD_s \tag{6}$$

$$IFSS = IFSS^* - 0.894 t_f CD_s \sigma_r \tag{7}$$

IFSS was the apparent interfacial shear strength, which combined the intrinsic interfacial shear strength IFSS* and the residual stress term. The shear strength at yield of the polymer substrate, $\tau_Y$, was estimated from their yield stress, $\sigma_Y$, using Von Mises equivalence ($\tau_Y = \sigma_Y/\sqrt{3}$) [31]. The yield stress of the PI substrate was equal to 68 MPa; the corresponding shear stress was equal to 39 MPa, which was much higher than the IFSS and IFSS*. This meant that delamination would occur at strain levels below the yield strain of the substrate. The interfacial shear strengths of ITO films with different thicknesses are listed in Table 2. IFSS* had no significant regularity with film thickness; that is, the interfacial strength was nearly independent of coating thickness. Considering residual compressive stress, IFSS of ITO with the same film thickness was higher than IFSS*. As the film thickness increased, the IFSS increased, leading the film to strip from the substrate with more difficulty. In other words, in a range of thickness in the ITO layer of 50–500 nm, the thicker ITO films delaminated under larger strains and had better adhesive properties where residual compressive stress had a positive effect.

In addition, it has been reported that the pretreatment method of substrate could improve the adhesion between the substrate and the film, such as plasma treatment, chromic acid treatment and the addition of primer. In follow-up works, it is necessary to study the comprehensive effects about residual stress and the pretreatment method of

substrate on the cohesive toughness and adhesion properties, to provide a theoretical basis for improving the mechanical durability of the films.

## 5. Conclusions

In this paper, ITO films on hard PC substrates were studied. In situ tests were carried out at a constant strain rate in the uniaxial mode on PC-ITO films with various film thicknesses to investigate the fracture behavior and electrical-mechanical properties of ITO films under tension.

The in situ test results show that the electrical failure strain and crack initiation strain decreased with increased film thickness; that is, the thinner ITO film had better ductility. Three fracture stages of ITO films under uniaxial tension were summarized from an in situ SEM test: I Crack initiation, II crack propagation, III crack saturation and delamination. Combining the mechanics model and the analysis of the three fracture stages under tension, as the film thickness increased (in the range of 50~500 nm), the fracture toughness decreased, whereas the interfacial shear strength increased. In addition, it is evident that internal stresses were to a considerable extent the practical ductility, cohesive toughness and adhesion property of ITO films on PC.

In conclusion, the mechanical durability of ITO films on PC substrates depended on the cohesive toughness (control cracking) and adhesive toughness (control delamination) where film thickness and residual stress had a non-negligible effect. The stability of ITO films on PC substrates could be expected to be improved by preparing ITO films (in the thickness ranging from 50 to 500 nm) with thinner thickness and higher residual compressive stress.

**Author Contributions:** Conceptualization, J.Z. and Y.Y.; methodology, X.Z. (Xuan Zhang) and X.Z. (Xiaofeng Zhang); investigation, J.Z., W.Z., J.L. and Y.C.; writing—original draft preparation, J.Z.; writing—review and editing, X.Z. (Xuan Zhang) and X.Z. (Xiaofeng Zhang); supervision, J.L.; project administration, W.Z.; funding acquisition, H.L. All authors have read and agreed to the published version of the manuscript.

**Funding:** This study is financially supported by the National Natural Science Foundation of China (No. 52072354).

**Institutional Review Board Statement:** Not applicable.

**Informed Consent Statement:** Not applicable.

**Data Availability Statement:** Not applicable.

**Conflicts of Interest:** The authors declare no conflict of interest.

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
