# Peer review of "Mechanical Properties of Tensile Cracking in Indium Tin Oxide Films on Polycarbonate Substrates"

_coatings, doi:10.3390/coatings12040538_

Round 1
Reviewer 1 Report
Authors in the current manuscript have investigated the influence of the mechanical stress on the mechanical and electrical properties of ITO film deposited on the thick polymer substrates. The have investigated the ITO stability as a function of external stress and the layer thickness. It is shown that the effective crack initiation strain decreases with increasing the layer thickness. But it also depends on the adhesion between the ITO layer and the substrate. This problem is roughly mentioned in the 4.3.2. The Interfacial Shear Strength but there is no solution proposed. Can author discuss with more details how the adhesion between PC and ITO can be enhanced, simultaneously the delamination decreased?
In general, it is a nice collection of important results but the missing information is how to improve the stability of the ITO on flexible substrate. Such discussion should be add to the manuscript. It is important for the practical application.
Moreover some points should be clarified:
How the substrate was prepared for the deposition? Did authors clean the surface before ITO deposition? If yes, what kind of cleaning was performed?
It should be in-situ and ex-situ instead of in situ and ex-situ
Line 100, should be … was measured by…. not was detected by ….
Line 209 it is see Figure 9 but in the manuscript are only 8 figures
Reviewer 2 Report
This paper describes experimental findings and understanding on the fracture behavior of ITO thin films deposited on Poly Carbonate substrate. As noted in this paper, mechanical properties of ITO thin films deposited on thin substrate have been well studied; however they are not very well studied in case when the substrate is thin (where Poisson's contraction/expansion of thin films is severely limited). This paper addresses one of the concerned properties, that is the fracture. Various experimental efforts lead to the conclusion that fracture of ITO is much easier to nucleate and grow with increase in the film thickness with the mechanics well described in the paper. The work presented in this paper is very well executed and the paper is well written. There are a couple of minor corrections to request for enhanced readability: 1) define the "vertical crack" better: the term "vertical crack" is somewhat confusing. Introduction of its definition in in early on, something like the "vertical crack developed in normal direction of the loading", may be helpful 2) the result shown in Fig.7 (R is not paralleled with crack density) may be understood from the frame of "percolation" theory. Discussion of it using the electron percolation may enable shorter and clearer description of the result.
